# The Recent Research Progress of NF-κB Signaling on the Proliferation, Migration, Invasion, Immune Escape and Drug Resistance of Glioblastoma

**DOI:** 10.3390/ijms241210337

**Published:** 2023-06-19

**Authors:** Pengfei Shi, Jie Xu, Hongjuan Cui

**Affiliations:** 1Cancer Center, Medical Research Institute, Southwest University, Chongqing 400716, China; spf2019@sina.cn (P.S.); xujiesw@sina.com (J.X.); 2Jinfeng Laboratory, Chongqing 401329, China; 3State Key Laboratory of Resource Insects, Southwest University, Chongqing 400716, China

**Keywords:** glioblastoma, NF-κB, proliferation, immune escape, drug resistance

## Abstract

Glioblastoma multiforme (GBM) is the most common and invasive primary central nervous system tumor in humans, accounting for approximately 45–50% of all primary brain tumors. How to conduct early diagnosis, targeted intervention, and prognostic evaluation of GBM, in order to improve the survival rate of glioblastoma patients, has always been an urgent clinical problem to be solved. Therefore, a deeper understanding of the molecular mechanisms underlying the occurrence and development of GBM is also needed. Like many other cancers, NF-κB signaling plays a crucial role in tumor growth and therapeutic resistance in GBM. However, the molecular mechanism underlying the high activity of NF-κB in GBM remains to be elucidated. This review aims to identify and summarize the NF-κB signaling involved in the recent pathogenesis of GBM, as well as basic therapy for GBM via NF-κB signaling.

## 1. Introduction

Glioblastoma multiforme (GBM) is the most frequent and aggressive form of primary central nervous system (CNS) tumor in humans [1]. Tumors are often located under the cerebral cortex, and most tumors grow throughout the cerebral hemispheres, showing invasive growth [2]. The main reasons for the poor prognosis of patients with GBM are the high degree of cellular heterogeneity and plasticity within the tumor, the infiltrating and migratory nature of GBM cells, and the high recurrence rate. Therefore, the 5-year survival rate of patients with GBM is only about 5%, and the median overall survival time is <14–16 months [3]. The occurrence of tumors also involves changes in the levels of many genes and proteins, as well as functional changes. One of the limiting factors affecting the treatment of GBM patients may be our lack of understanding of the underlying mechanisms of GBM development in vivo. Therefore, it is necessary to further explore the growth mechanism of GBM for the treatment and prolonging of the survival time of patients.

Nuclear factor-κB (NF-κB) contains a family of five transcription factors that form unique protein complexes that bind to common DNA sequences in the promoter regions of response genes that regulate cellular processes [4]. The past few decades have witnessed significant progress in understanding the NF-κB signaling pathway under physiological and pathological conditions [5]. The role of NF-κB in the occurrence, development, metastasis, and treatment resistance of human cancer has attracted special attention [6,7]. Because of the inflammatory microenvironment and various carcinogenic mutations, a large number of human cancers have constitutive NF-κB activity. The activity of NF-κB not only promotes tumor cell proliferation, inhibits cell apoptosis, and attracts angiogenesis, but also induces epithelial–mesenchymal transition (EMT) and promotes distant metastasis [8,9]. In some cases, NF-κB activation may also reshape local metabolism, making the immune system powerless, thereby promoting tumor growth [10]. The inhibition of NF-κB in tumor cells usually leads to tumor regression, making the NF-κB pathway a promising therapeutic target [11]. Therefore, the purpose of this review is to summarize the molecular mechanisms and treatment methods of GBM progression that have been revealed from the perspective of NF-κB signaling in the past two years. Although there are many small molecule compounds or traditional Chinese medicine monomers that can inhibit the growth of GBM against NF-κB signals, there are few clinical trials. Therefore, it is also hoped that this review can provide some guidance for targeted therapy and clinical applications of GBM.

## 2. NF-κB

NF-κB, a rapidly inducible transcription factor, has five members: RelA (p65), RelB, c-Rel, NF-κB1, and NF-κB2 [12,13] (Figure 1A). In contrast to other family members, NF-κB1 and NF-κB2 precursor forms exist (p105 and p100) and are ultimately hydrolyzed to produce functional proteins (p50 and p52) [14,15]. All five members of this protein family can form homologous or hetero dimers (the main physiological function within the cell is the p50–p65 dimer) [16,17], and share some structural characteristics, including a Rel homologous domain (RHD), which is necessary for dimerization and binding to homologous DNA elements [18]. In most resting cells, protein IκB family members (NF-κB inhibitor) can bind to these dimers and inhibit their function. The main members of the IκB family are IκBα, IκBβ, IκBɛ, and Bcl-3 [19,20] (Figure 1B). These members are characterized by ankyrin repeats, which bind to the DNA-binding domains of transcription factors, thereby rendering them transcriptionally inactive. By structural analysis, it was found that p105 and p100 also contain ankyrin repeats and, consequently, are cleaved to p50 and p52 upon maturation [21]. When p50 or p52 bind to members containing transactivation domains, such as p65 or RelB, they constitute transcriptional activators.

In general, NF-κB activation occurs through the release from IκB molecules, or maturation and division of p100 and p105. A prerequisite for dependence on IκB degradation is that IκB is pre-phosphorylated by IκB kinases (IKKs) in two key N-terminal sequences [23]. IKK activity is present in a large protein complex that contains two catalytic subunits, IKKα and IKKβ, and a stent subunit, IKKγ [24] (Figure 1C). Activators of the IKK complexes include mitogen-activated protein kinase kinases (MAP3Ks), such as mitogen-activated protein kinase 1/3 (MEKK1/3) and transforming growth factor beta-activated kinase 1 (TAK1). TAK1 is a member of a larger protein kinase complex comprising TAK1, TAB1, and TAB2, which phosphorylates IKK2 and NIK [24,25,26]. MEKK3, a member of the MAP3K family, is known to play a role in TLR4-mediated signal transduction and regulate the activation of c-Jun and NF-κB in endothelial inflammation [27]. Overall, two main NF-κB-activating pathways exist in cells. The canonical pathway is induced by most physiological NF-κB stimuli. These stimuli come from many sources, including Toll-like receptors (TLRs), the interleukin 1 (IL-1) receptor (IL-1R), the tumor necrosis factor (TNF) ligand and receptor (TNFR) superfamily, and ligands for B- and T-cell receptors [28,29,30,31,32].

The canonical NF-κB signaling pathway mainly relies on IKKβ and NEMO, leading to phosphorylate IκBα on serine residues S32 and S36, which results in polyubiquitination and the subsequent degradation of IκB by 26S proteasomes [33,34] (Figure 2). Subsequently, the complex releases NF-κB (mainly the p50–p65 heterodimer), and the p50–p65 heterodimer translocates to the nucleus, binds the κB site, and activates gene transcription [35]. The intracellular signaling components mediating non-canonical NF-κB activation differ significantly from those involved in canonical NF-κB activation. In the non-canonical NF-κB signaling pathway, activation of receptor activators for nuclear factor kappa B (RANK), B-cell activation factor (BAFFR), lymphtoxin β-receptor (LTβR), or CD40, results in the activation of IKKα by the NIK [36]. The activation of IKKα can further the phosphorylate p100 on serine residues S870 and S866 [37,38]. The phosphorylation of p100 can lead to the polyubiquitination and degradation of p52 [39,40].

As the most common subunits of p65 and p50, they were found to be inactive in unstimulated cells, mainly in the cytoplasm, complexed with members of the IκB inhibitor protein family. In this composite form, the nuclear localization signal (NLS) found on the subunit of heterodimer p50–p65 is identified by IκBα, the most predominant IκB, which prevents nuclear translocation, DNA binding, and the subsequent activation of target gene transcription [41]. However, IL-1 or TNFR activation induces the activation of the IκB kinase complex, which phosphorylates IκBα, causing the ubiquitination and destruction of the phosphorylated IκBα protein by the proteasome [24]. As a consequence of IκBα degradation and NLS unmasking, the majority of NF-κB complexes translocate to the nucleus within minutes, where they bind cognate DNA sites and regulate gene transcription. In many cases, NF-κB dimer activation of target genes requires the assistance of other transcription factors, including STAT, AP1 family members, p53, NRF2, IRFs, etc. In addition, some transcription factors were also found in the NF-κB-binding DNA region, such as FOXM1, AP1, ZBTB7a, E2A, STAT1, IRF, ATF2, RPS3, E2F1, BCL6, and KLF6 [42]. Further validation is needed to determine whether these factors are involved in the transcription of NF-κB and how they are involved.

The NF-κB dimer can directly transcribe approximately 1700 genes with the help of multiple cofactors, and, based on the function of genes, they are divided into the following parts: cell adhesion molecules (such as CD44 and CD59), cell surface receptors (such as ADAM12 and LDLR), cytokines/chemokines and their modulators (such as CCL4 and IL4), enzymes (such as FUT4 and USP38), growth factors, ligands, and their modulators (such as EGFR and HGF), immunoreceptors (such as CD14 and CD33), miscellaneous (such as BRAF and CHD1), proteins involved in antigen presentation (such as PSMB9 and MGEA5), regulators of apoptosis (such as BAD and BAX), stress response genes ( such as COX2 and GADD45A), transcription factors and their modulators (such as ATF3 and BCL6), and miRNA (such as miR-223 and miR-34A) [43]. It can be seen that NF-κB affects cell growth by regulating different physiological processes.

## 3. NF-κB Activation Is Involved in Development and Progression of Glioblastoma

NF-κB family members and their regulatory genes are associated with glioblastoma cell malignant transformation, proliferation, survival, angiogenesis, invasion/metastasis, and therapeutic resistance [44]. This review mainly summarizes the role of NF-κB in the occurrence and development of glioblastoma in the past two years. Based on the focus of the article and the direction of the researchers, we divide it into five parts: inflammation and promotion, migration and invasion, immune escape, drug resistance, and apoptosis.

### 3.1. Inflammation and Proliferation

NF-κB is widely used by eukaryotic cells as a gene regulator to control cell proliferation and cell survival. Recent research progress suggests that some genes can promote the inflammation and proliferation of GBM cells through the activation of NF-κB signaling. For example, sphingosine kinase 1 (SPHK1) promoted inflammation through the NF-κB/IL-6/STAT3 signaling pathway, which leads to the phosphorylation of JNK. Pentraxin 3(PTX3), activated by the JNK-JUN and JNK-ATF3 pathways, can interact with SPHK1 to form a positive feedback loop to reciprocally increase expression, and promote inflammation and GBM growth [45]. Orexin-A (OXA, a neuropeptide) and YTHN6-methyladenosine RNA-binding protein 2 (YTHDF2) promote GBM cell inflammation via activating the NF-κB signaling pathway [46,47]. In addition, some recent studies have found that activation of NF-κB signaling promotes cell proliferation through the ubiquitination of factors of the NF-κB pathway. For instance, tripartite motif-containing 25 (TRIM25), an essential E3 ubiquitin ligase, can accelerate the malignant progression of GBM through NF-κB activation [48]. Tri-domain protein 22 (TRIM22) was first identified as an IFN-induced protein and was found to be a transcription target gene for TP53 [49]. TRIM22 contains a conserved cyclic domain, indicating that it may participate in the post-transcriptional modification of certain proteins as an E3 Ub ligase [50]. In GBM cells, TRIM22 can bind to IκBα and accelerate its degradation by inducing ubiquitination of the K48 site. TRIM22 also formed a complex with IKKγ and promoted K63-linked ubiquitination, which results in the phosphorylation of both IKKα/β and IκBα [51]. In addition, TRIM22 forms a complex with cytosolic purine 5-nucleotidase (NT5C2) and promotes K63-linked ubiquitination of retinoic acid-inducible gene I (RIG-I), which activates NF-κB/CCAR1 (cell division cycle and apoptosis regulator 1) to induce cell proliferation of GBM [52]. This result shows that E3 Ub ligase induces NF-κB signaling in GBM, driving tumor growth and progression.

In addition to the research on the factors directly affecting the NF-κB signaling pathway, some studies have shown through simple analysis that the cell proliferation of GBM is regulated by NF-κB signaling. For instance, RNA-binding protein TAR DNA-binding protein 43 (TDP43) and transcription factor activator protein 1 (AP-1) can upregulate circADAMTS6, and circADAMTS6 is specifically upregulated in the hypoxic microenvironment of GBM, and can also accelerate GBM progression via the ANXA2/NF-κB pathway [53]. Methyltransferase-like 3 (Mettl3) enhances the stability of metastasis-associated lung adenocarcinoma transcript 1 (MALAT1) and activates NF-κB with the assistance of HuR to promote the malignant progression of GBM [54]. Chitinase-3-like protein 1 (CHI3L1) binds to actinin alpha 4 (ACTN4) and NF-κB1 and enhances NF-κB signaling by promoting p65 nuclear translocation to induce GBM progression [55]. Crumbs homolog 2 (CRB2) upregulates the tumor necrosis factor α (TNFα)/NF-κB signal to promote the malignant progression of GBM [56]. NIMA-related kinase 2 (NEK2), a member of the NIMA-related kinase family, activates the NF-κB signaling pathway by phosphorylating NIK and increasing its activity and stability, promoting the malignant progression of GBM [57]. Fatty acid-binding protein 5 (FABP5) encourages GBM cell proliferation via raising NF-κB signaling [58]. Gold nanoparticles (AuNPs) have become a promising cancer treatment nanomaterial, which can inhibit the TRAF6/NF-κB pathway to reduce the growth of GBM cells [59]. NF-κB inhibitor BAY 11-7821 can inhibit GBM cell proliferation by downregulating the NF-κB signaling pathway [60]. BX795, an inhibitor of TBK1, can decrease the expression of NIK, IKK, and TNFα to suppress GBM cell proliferation [61]. RAS-selective lethal 3 (RSL3), a well-known inhibitor of glutathione peroxidase 4 (GPX4), can activate NF-κB signaling and inhibit the expression of downstream genes ATF4 and xCT, which prevents iron death, induced by GPX4 expression to facilitate cell proliferation [62].

Some of the studies here indirectly indicate that the gene regulates GBM cell proliferation through phosphorylation, nuclear translocation, and the stability of p65. For example, TCF4N, an isoform of the β-catenin interacting transcription factor TCF7L2, binds and promotes s536 phosphorylation, nuclear translocation, and the stability of p65, ultimately upregulating the NF-κB target gene in GBM cells [63]. G protein inhibitory α subunit 2 (Gαi2) increases the expression of the NF-κB target gene Sp1 (a transcription factor) through dephosphorylation of the p65 subunit, and at the same time, Sp1 has been confirmed to bind to the Gαi2 promoter region, mediating Gαi2 overexpression [64]. Calponin 3 (CNN3), melanoma cell adhesion molecule (MCAM, also named CD146), chemerin (also known as retinoic acid receptor responder protein 2 (RARRES2)), retinol-binding protein 1 (RBP1), growth differentiation factor 15 (GDF15), G protein-coupled receptor 17 (GPR17), and cysteamine (2-aminoethanethiol) dioxygenase (ADO) can induce the phosphorylation of p65 to promote GBM cell proliferation [65,66,67,68,69,70,71].

There are also some molecular mechanisms that interpret GBM proliferation through NF-κB signals via non-coding RNA [72,73]. LncRNA-PRADX can increase the trimethylation of H3K27 in the UBXN1 gene promoter through polycomb repressive complex 2 (PRC2)/DEAD box protein 5 (DDX5) complex recruitment and promotes NF-κB activity [74]. The SWI/SNF complex antagonist associated with prostate cancer 1 (SChLAP1, a lncRNA) is increased in primary GBM samples and cell lines and enhances and stabilizes interaction with the protein actinin alpha 4 (ACTN4), which leads to increased nuclear localization of the p65 and activation of NF-κB signaling [75]. LINC01057 interacts with IKKα and maintains IKKα nucleus localization, leading to the activation of NF-κB signaling in GBM cells [76]. These recent research advancements have enriched the molecular mechanisms by which the NF-κB signaling pathway regulates GBM cell proliferation, and also provided the possibility of targeting NF-κB signaling for GBM treatment.

### 3.2. Migration and Invasion

The invasiveness of tumor development mainly depends on the complex biochemical and biological changes in the tumor cells themselves and their related matrices. Recent research progress suggests that the change of NF-κB signaling can assist GBM cell migration and invasion. For example, TNF-weak-like factor (TWEAK) can upregulate the nuclear E2F4 and E2F5 protein levels in GBM cells, and E2F4 and E2F5 can bind to the promoter region of NIK to regulate the transcription and expression of NIK, thereby activating the NF-κB signaling pathway to promote cell invasion [77]. In addition, the downstream gene fibroblast growth factor-inducible 14 (Fn14) of TWEAK can make GBM cells more invasive and lethal by inducing p50 expression [78]. Esophageal carcinoma-related gene 4 (ECRG4) can inhibit AKT/GSK3β/β-catenin signaling to reduce tumor cell migration and invasion [79], however, ubiquitin protein ligase E3 module N-recognition 5 (UBR5) can bind to and degrade ECRG4, thereby promoting the phosphorylation and nucleation of p65, and inducing the migration and invasion of GBM [80]. Ras-like without CAAX1 (RIT1), a member of the Ras family, promotes GBM cell invasion via the AKT/ERK/NF-ĸB signaling pathway [81]. Transforming growth factor-β (TGF-β) signaling can upregulate the expression and nuclear transport of claudin-4 (CLDN4). CLDN4 promotes the epithelial–mesenchymal transition of GBM by modulating tumor necrosis factor-α (TNF-α)/p-IKKα/p-p65 signals [82]. Fos-like antigen 1 (FOSL1) can promote the pre-neural mesenchymal transformation of glioblastoma stem cells through the NF-κB signaling pathway [83]. The expression of breast cancer metastasis suppressor 1 (BRMS1) in mut p53 GBM aggravates patient prognosis and promotes cell migration and invasion by activating EGFR-AKT-NF-κB signaling [84]. Additionally, EGFR signaling also promotes GBM cell invasion via activation of the TAB1-TAK1-NF-κB-EMP1 pathway [85] and induces the expression of minichromosome maintenance 8 (MCM8) to maintain the clonogenic and tumorigenic potential of GSCs [86]. However, beyond that, the GLIS family’s zinc finger 3 (GLIS3), glutathione peroxidase 8 (GPX8), and zinc transporter ZIP7 (SLC39A7) encourage GBM cell migration and invasion by activating NF-κB signaling [87,88,89]. Annexin-A1 (ANXA1), hypoxia-induced procollagen lysyl hydroxylase 1 (PLOD1), and cell surface glucose-regulated protein 78 (csGRP78) can enhance the phosphorylation of p65 to induce GBM cell migration and invasion [90,91,92].

Moreso, some non-coding RNA is also involved in GBM migration and invasion. For instance, circKPNB1 regulates the protein stability and nuclear translocation of SPI1 (a member of the ETS transcription factor family) [93]. SPI1 can promote the malignant phenotype of GBM stem cells via tumor necrosis factor (TNF)-alpha mediated NF-κB signaling. SPI1 can maintain the stability of circKPNB1 by transcriptionally upregulating DGCR8 expression and forming a positive feedback loop among SPI1, circKPNB1, and DGCR8 [93]. LINC01393 sponged miR-128-3p to upregulate nucleolar and spindle-associated protein 1 (NUSAP1), which results in the development and progression of GBM via activating the NF-κB pathway [94]. Dipotassium glycyrrhizinate (DPG), a dipotassium salt of glycyrrhizic acid (isolated from licorice) can upregulate the expression of miR-4443 and miR-3620, which decrease the RNA levels of the downstream target genes CD209 and TNC of NF-κB and inhibits the migration of GBM [95]. However, recent research on some non-coding RNA can also inhibit the migration and invasion of GBM cells by inhibiting NF-κB signaling. MiRNA-451 can target IKKβ to regulate the NF-κB signaling pathway and inhibit the growth of GBM cells [96]. MiR-19a/b can target septin7 (SEPT7) to reduce its expression, thereby upregulating the expression of p-AKT and p65 to promote the migration and invasion of GBM [97]. LncRNA, activated by TGF-β (lncRNA-ATB) encourages P65 translocation into the nucleus, thus facilitating GBM cell invasion [98]. LINC00526 can repress PI3K/AKT/NF-κB signaling to inhibit GBM cell migration and invasion [99].

Meanwhile, some recent studies have also hindered the migration and invasion of GBM cells by inhibiting NF-κB signaling. The case in point is bisdemethoxycurcumin (BDMC), from the rhizome of turmeric (Curcuma longa), which significantly reduces protein levels associated with the PI3K/AKT, Ras/MEK/ERK pathways, which reduces the expression of NF-κB, MMP-2, MMP-9, and N-cadherin, thereby inhibiting cell migration and invasion of GBM [100]. In addition, curcumin also induces tumor cell apoptosis by suppressing the heat shock protein 60 (HSP60)/Toll-like receptor 4 (TLR-4)/myeloid differentiation primary response 88 (MYD88)/NF-κB pathway [101]. Eriodictyol, a natural flavonoid, can suppress GBM cell migration and invasion by downregulating the PI3K/AKT/NF-κB pathway [102]. Tat-NTS is a synthesized small molecular peptide that represses annexin-A1 (ANXA1) nuclear translocation, and it can suppress GBM cell migration and invasion by diminishing the phosphorylation level of p65 [103]. Fentanyl can inhibit p65 activation to decrease the invasiveness of GBM cells [104].

### 3.3. Immune Escape

When the body undergoes an inflammatory response, it initiates a natural immune response. These natural immune response cells are able to release inflammatory factors, and the transcription of these cytokines depends on the activation of NF-κB signaling [105]. Therefore, the NF-κB signal regulates the adaptive immunity of tumors and can promote the immune escape of GBM cells. Compared to the progress in proliferation, migration, and invasion, there is relatively little research progress on immune escape. For example, the T-cell immunoglobulin domain and mucin domain protein 3 (TIM-3, an immune checkpoint) is mainly expressed in immune cells, regulating innate and acquired immunity [106]. In GBM cells, TIM-3 can activate the phosphorylation of NF-κB, which in turn stimulates IL-6 secretion and STAT3 phosphorylation to promote cell growth [107]. Annexin-A1 (ANXA1), a calcium-dependent phospholipid-binding protein, activates p65 by interacting with the IKK complex, promoting the binding of p65 dimers to specific promoter regions of interleukin-8 (IL-8), and mediating immune escape of GBM [108]. Polymerase 1 and transcript release factor (PTRF/Cavin-1) interacts with lncRNANEAT1 and stabilizes the mRNA of nuclear paraspeckle assembly transcript 1 (NEAT1). PTRF also promotes the activity of NF-κB by inhibiting the expression of UBX domain protein 1 (UBXN1) via NEAT1 and enhancing the transcription of PD-L1 through NF-κB activation. Thus, PTRF promotes the immune evasion of GBM cells by regulating PD-1 binding and PD-L1-mediated T-cell cytotoxicity [109].

In addition, the Warburg effect endows tumor cells with the ability to utilize aerobic glycolysis to meet their high metabolic needs [110]. However, under different concentrations of sugar, there are different ways of regulation within the cells. Glycogen branching enzyme 1 (GBE1) will decrease the expression of fructose-bisphosphatase 1 (FBP1) via p65 activation, which enhances the Warburg effect to drive GBM progression [111]. High glucose promotes the dissociation of hexokinase (HK) 2 from mitochondria and subsequently binds and phosphorylates at T291 of IκBα. Additionally, high glucose also induces the expression of IκB kinase α/β (IKKα/β) and NIK [112]. This leads to increased interaction between IκBα and μ-calpain protease and promotes the degradation of IκBα by μ-calpain protease, which activates NF-κB signaling to increase the immune escape of GBM [113]. Glutamate dehydrogenase 1 (GDH1) is a key enzyme in the decomposition of glutamine, converting glutamic acid into α-ketoglutaric acid (α-KG). In addition, under low sugar conditions, GDH1 is phosphorylated at serine 384 and interacts with RelA and IKKβ, and α-KG directly binds and activates IKKβ and NF-κB signaling, which promotes glucose uptake and tumor cell survival by upregulating glucose transporter 1 (GLUT1), thereby accelerating the formation of GBM [114].

### 3.4. Drug Resistance

Glioblastoma is a highly fatal brain cancer with a high recurrence rate [115,116]. In the past decade, treatment progress has been slow, and the prognosis has been extremely poor [3]. Temozolomide (TMZ) is a new oral alkylating agent anti-tumor drug with broad-spectrum anti-tumor activity. It can pass through the blood–brain barrier, and its bioavailability is close to 100% [3]. It has also been used clinically, and can effectively treat newly diagnosed and relapsed glioblastomas and anaplastic astrocytomas, prolong the survival period of patients, and has good safety and tolerance [117]. However, overall, the efficacy of TMZ does not satisfy the majority of patients. With the use of drugs, patients develop resistance, resulting in both TMZ and targeted drugs becoming ineffective and thus no longer having any anti-tumor effects. Therefore, it is with urgency that we focus on the molecular mechanisms of drug resistance and the therapy of drug resistance.

#### 3.4.1. Promoting Resistance

Recent studies, from the perspectives of NF-κB signaling and TMZ, have elucidated the molecular mechanism of GBM resistance. For example, the expression of RNA-binding protein ADAR3 can activate NF-κB signaling by altering the expression or activity of one of the many upstream factors that regulate NF-κB activity to increase the GBM cell resistance for TMZ, and NF-κB signaling also promotes the expression of ADAR3 [118]. The WD repeat-containing protein 5 (WDR5)/mixed lineage leukemia (MLL) complex can bind to the promoter of E3-ligases FBXO32, mediating active histone modifications, including H3K4me3 and H3K9ac, leading to the upregulation of FBXO32, and further activated NF-κB/O(6)-methylguanine-DNA methyltransferase (MGMT) signaling via the ubiquitin-dependent degradation of IκBα, which further downregulates the expression of brain acid soluble protein 1 (BASP1) to promote the resistance of GBM cells to TMZ [119]. Tryptophan hydroxylase 1 (TPH-1) improves the production of serotonin in GBM cells. Then, the increased serotonin enhances the NF-κB signaling pathway by upregulating the L1-cell adhesion molecule (L1CAM), thereby promoting drug resistance [120]. Enhancer of zeste homolog 2 (EZH2, a histone-lysine N-methyltransferase) can promote TMZ resistance by activating NF-κB signaling [121]. Calpain-2 can downregulate DNA damage signaling proteins, such as TP53 and TP53BP1, to block DNA damage recognition and active NF-κB signaling to inhibit the sensitivity of GBM cells to TMZ [122]. Mucin1 (MUC1), by regulated NF-κB signal transcription, can promote GBM progression and TMZ resistance by stabilizing EGFRvIII [123]. Additionally, the EGFRvIII activation NF-κB pathway further regulates the expression of ALDH1A3, promoting the anterior neural mesenchymal transformation of GBM, thus reducing its sensitivity to TMZ [124].

There are also studies that elucidate the drug resistance mechanism of GBM from the perspective of p65 phosphorylation. For instance, receptor-interacting protein 2 (RIP2) promotes SOX2 expression by upregulating the phosphorylation of p65 to enhance GBM cell stemness and resistance for TMZ [125]. In addition, RIP2 and pleckstrin homology containing family member 5 (PLEKHG5) also induce the expression of O6-methylguanine-DNA methyltransferase (MGMT) by enhancing the phosphorylation of p65 to promote GBM cell resistance for TMZ [126,127]. A-kinase-interacting protein 1 (AKIP1) promotes radiotherapy tolerance of GBM by regulating C-X-C Motif Chemokine Ligand 1/8 (CXCL1 and CXCL8)-mediated p65 phosphorylation [128].

Furthermore, the transition from PN to MES is considered a marker of GBM recurrence and resistance to multiple therapies. For instance, lipopolysaccharide (LPS)-induced tumor necrosis factor (TNF)-α factor (LITAF, also named PIG7, a translation factor) promotes target genes (including TNF-α, IL-6, CCL-2, etc.), thereby further activating the NF-κB signal to promote a mesenchymal transition and increase the radiosensitivity of GBM [129]. Actin-related protein 2/3 complex subunit 1B (ARPC1B) can activate the NF-κB and STAT3 signaling pathways by inhibiting the TRIM21-mediated degradation of IFI16 and HuR, which promotes MES phenotype maintenance and radiotherapy resistance of GBM [130].

#### 3.4.2. Inhibiting Resistance

Tumor cell resistance can be divided into two categories: intrinsic (existing when not in contact with drugs) and acquired (produced after contact with drugs) [131]. Generally speaking, developing resistance to an anti-tumor drug may result in cross-resistance to structurally and functionally similar drugs, while it remains sensitive to other non-identical drugs. Therefore, one of the methods for treating GBM resistance is drug rotation [132]. Sequential chemotherapy can be used as an adjuvant therapy, which can, to some extent, restore the sensitivity of drug-resistant tumor cells to therapeutic drugs, and can also be considered as a disguised delay in tumor resistance [133]. In addition, combining traditional Chinese medicine or anti-angiogenic drugs is also an effective way to treat GBM resistance [134]. Interestingly, recent studies have mostly combined traditional Chinese medicine ingredients to treat GBM resistance through the NF-кB signaling pathway.

For example, guggulsterone, a major active steroid extracted from myrrh, has been found to inhibit cancer cell growth [135]. Guggulsterone can downregulate EGFR/PI3K/AKT signaling and the NF-кB pathway, and enhance TMZ-induced GBM growth inhibition and apoptosis [136]. Hesperetin (HSP), derived from citrus fruits, can inhibit PI3K/AKT and NF-κB pathways to induce GBM cell apoptosis and suppress metastasis [137]. Phenethyl isothiocyanate (PEITC), extracted from cruciferous vegetables, significantly reduced the levels of proinflammatory cytokines, such as IL-1β, TNF-α, and IL-6, in transcriptional levels and modulated AKT- and ERK-dependent and NF-*κ*B signaling pathways in GBM [138]. Apigenin can inhibit the p65/HIF-1α-mediated expression of glucose transporter 3 (GLUT3) and pyruvate kinase isozyme-type M2 (PKM2) and wakens the stem cell and DNA damage repair of GBM cells, thereby increasing the radiosensitivity of GBM [139] (Table 1).

In addition, some traditional Chinese medicine ingredients clearly regulate GBM resistance through p65. For instance, cynaropicrin (CYN), a natural compound that was isolated from an edible plant (artichoke), can induce ERK dephosphorylation accompanied by a reduction of p65 to promote GBM cell apoptosis via inhibiting the p62/Keap1/Nrf2 pathway; additionally, CYN also increases the cytotoxicity of TMZ to GBM [140]. Lycorine, an isoquinoline alkaloid isolated from lycoris, can induce the production of reactive oxygen species (ROS) and the downregulation of the phosphorylation of p65 to inhibit GBM cell proliferation and enhance the therapeutic effect of TMZ [141]. Tubeimoside-I (TBMS1), a saponin from traditional Chinese medicine, can downregulate the expression of p-PI3K, p-AKT (Ser473), p-mTOR (Ser2481), and p-p65(Ser536) in GBM cells, and enhance the therapeutic effect of TMZ [142]. Regorafenib, the oral multi-kinase inhibitor, can make GBM sensitive to TMZ treatment by upregulating the expression of CXCL12, CXCR4, p-ERK, and p-p65 [143]. Rabeprazole, extracted from Aciphex tablets, can reduce the resistance of GBM to TMZ by repressing EMT via weakening p65 activation and IκBα phosphorylation [144]. 

### 3.5. Apoptosis

In addition to inhibiting GBM resistance, some studies have also found that traditional Chinese medicine can directly inhibit NF-κB signaling and promote cell apoptosis. The case in point is that myrislignan inhibits the activation of NF-κB signals by blocking the phosphorylation of p65 proteins, and induces iron death in GBM cells through the Slug-SLC7A11 signal pathway [145]. Monacolin K (MK), a polyketo secondary metabolic compound of the mold genus monascus, can downregulate JNK/ERK/P65/IκBα expression and promote the phosphorylation of ERK and p65, ultimately resulting in the apoptosis of GBM cells [146]. Poly(allylamine hydrochloride) (PAH) can induce GBM cell apoptosis via suppressing TGF-β/p65 signaling [147]. Meisoindigo, a second-generation derivative of indirubin, can diminish the expression of PI3K, AKT, p-AKT, p65, and p-p65 to result in apoptosis of GBM cells [148]. Furthermore, cannabidiol (CBD, from cannabis sativa plants), lactucopicrin (LCTP, a natural sesquiterpene lactone from Lactucavirosa), Rosmarinic acid (RA, a Fyn kinase inhibitor), and remimazolan can induce cell apoptosis of GBM via repressing NF-κB signaling [149,150,151].

In addition, the expression of LncRNA can also induce apoptosis in GBM cells by downregulating NF-κB signaling. LncRNA-DRAIC can interact with the IKK complex, specifically with IKKα and NEMO, and damage the integrity of the complex, thereby repressing IκBα phosphorylation and the NF-κB signaling pathway to induce GBM cell apoptosis [152]. LncRNA IGF1R antisense imprinted non-protein coding RNA (IRAIN) inactivated the IGF-1R/PI3K/NF-κB signaling to repress GBM development [153].

## 4. Summary and Prospects

NF-κB is a central factor in inflammation, stress responses, cell differentiation, or proliferation, and cell death. It can be activated by various stimuli and complex signal pathway networks, and these signal pathways can also interact with each other. In the past two years, we have gained new insights into the function of NF-κB signaling in GBM inflammation, proliferation, migration, migration, immune escape, drug resistance, and apoptosis. In these studies, it is not difficult to find that most of the progress of GBM resolution is achieved by regulating EGFR/PI3KAKT/ERK/NF-κB signaling, with NF-κB signaling mainly characterized by p65 phosphorylation, stability, and nuclear transport. This further indicates that we are gradually mastering the signal network that regulates GBM growth through NF-κB signaling. However, these studies mostly demonstrate the ability of this gene to regulate GBM cell proliferation, migration, and invasion through the activation of intracellular NF-κB signals, as well as experiments on proliferation and immune escape phenomena. Such studies have not demonstrated that this gene can directly bind or participate in the modification or expression of various factors of the NF-κB signaling pathway. Secondly, there is no detailed explanation of why activation of the NF-κB pathway promotes cell proliferation, migration and invasion, immune escape, and drug resistance, and why the inhibition of the NF-κB pathway can induce cell apoptosis. These studies lack a regulatory network for downstream target genes of NF-κB, therefore, in future research, we need to increase the integrity of the experiment and increase the expression and molecular regulatory networks of downstream target genes of NF-κB signaling to explain the proliferation, migration and invasion, immune escape, or drug resistance of GBM cells, rather than relying on some phenomenon experiments to conclude that this gene regulates GBM cell proliferation, migration and invasion, or immune escape through NF-κB signaling.

In addition to understanding the molecular mechanisms of carcinogenic effects, the true value of elucidating signal transduction pathways is the clinical application of this knowledge. Therefore, the NF-κB signaling pathway is an attractive target for therapeutic interventions. Pharmacological inhibition can be achieved by acting on the upstream of IKK (ligands, receptors, and adapter proteins) and IKK levels (inhibiting IKK, IκΒ stabilization) and downstream of IKK (p65 nuclear translocation, DNA binding). Generally speaking, NF-κB-targeted therapy includes biomolecular inhibitors, natural compounds, and synthetic molecules. However, recent research mainly focuses on inducing GBM cell apoptosis with natural compounds or treating GBM cell resistance with the chemotherapy drug TMZ. Although there are many small molecule compounds or traditional Chinese medicine monomers that can inhibit the growth of tumors against NF-κB signals, there are few clinical trials in GBM. Therefore, we can utilize these research advancements to design targeted drugs and peptides and combine these natural or small molecule compounds for clinical therapy of GBM. When designing therapy methods, consideration should be given to the possible serious and systemic side effects of NF-κB, such as widespread immune suppression. Perhaps, NF-κB inhibitors will be used as adjunctive therapy in combination with chemotherapy/radiotherapy. In conclusion, the lack of clinical trials emphasizes the necessity of conducting such studies.

## Figures and Tables

**Figure 1 ijms-24-10337-f001:**
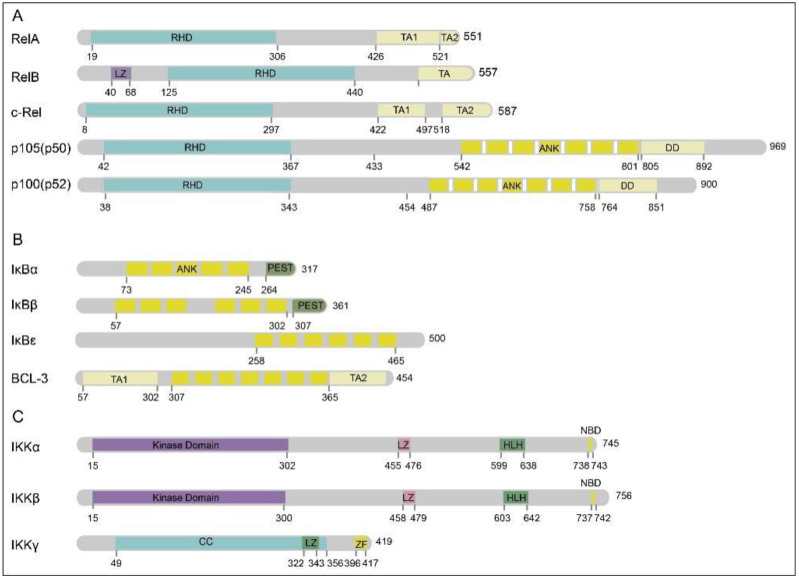
The core members of the NF-κB signaling pathway [22]. (**A**) The protein structure of the core member of the NF-κB signaling; (**B**) the protein structure of the core member of the IκB family; (**C**) the protein structure of the core member of the IκB kinase (IKK) complex. Further abbreviations: N-terminal Rel homology domain (RHD); ankyrin (ANK); C-terminal transactivation domains (TAs); leucine zipper-like motif (LZ); death domain (DD); helix–loop–helix domain (HLH); coiled-coil domain (CC); NEMO-binding domain (NBD); zinc finger domain (ZF).

**Figure 2 ijms-24-10337-f002:**
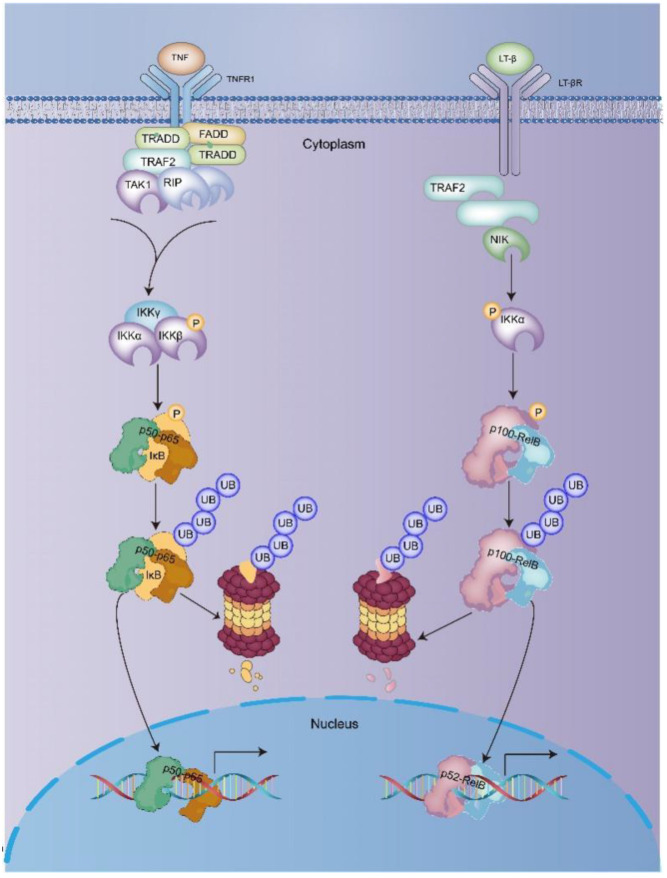
Activation of NF-κB signaling [19]. Further abbreviations: tumor necrosis factor receptor 1 (TNFR1)-associated death domain protein (TRADD); Fas-associated protein with death domain (FADD); tumor necrosis factor (TNF) receptor-associated factor 2 (TRAF2).

**Table 1 ijms-24-10337-t001:** Potential drugs for the treatment of glioblastoma via NF-κB signaling.

Drugs	Source	Testing Index	Efficacy
Guggulsterone [136]	Myrrh	EGFR; PI3K; AKT; p-p65	Inhibiting drug resistance
Hesperetin [137]	Citrus fruits	PI3K; AKT; p-AKT; p65	Inhibiting proliferation; Inducing apoptosis; suppressing metastasis
Phenethyl isothiocyanate [138]	Cruciferous vegetables	PI3K; p-AKT;p-p65; p-IKKα/β; IKKα/β	Inhibiting drug resistance
Apigenin [139]	Celery	p65	Improving sensitivity to radiotherapy
Cynaropicrin [140]	Cynara scolymus	ERK; p65	Promoting the pharmacodynamics of TMZ
Lycorine [141]	Lycoris	p-p65	Inhibiting proliferation; reducing drug resistance
Tubeimoside-I [142]	Bolbostemma paniculatum	EGFR; p-PI3K; p-AKT; p-mTOR; p-p65	Promoting the pharmacodynamics of TMZ
Regorafenib [143]	Compound	p-ERK; p-p65	Reducing drug resistance
Rabeprazole [144]	Aciphex tablets	p-p65; p- IκBα	Reducing drug resistance
Myrislignan [145]	Myristica fragrans	p-p65	Inducing ferroptosis
Monacolin K [146]	Compound	JNK; ERK; p65; IκBα	Inducing apoptosis
Allylamine hydrochloride [147]	Biomaterials	p-65	Inducing apoptosis
Meisoindigo [148]	Compound	PI3K; AKT; p-AKT; p65; p-p65	Inducing apoptosis
Cannabidiol [149]	Cannabis sativa plants	p65; p-p65	Antioxidative; Inducing apoptosis
Lactucopicrin [150]	Lactucavirosa	p65	Promoting autophagy; inhibiting proliferation; oxidative stress
Rosmarinic acid [151]	Rosemary	PI3K; AKT; p-AKT; p65	Inhibiting proliferation; suppressing migration and invasion; inducing apoptosis

## Data Availability

Not applicable.

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
