# Peer review of "The Recent Research Progress of NF-κB Signaling on the Proliferation, Migration, Invasion, Immune Escape and Drug Resistance of Glioblastoma"

_ijms, 2023, doi:10.3390/ijms241210337_

Round 1
Reviewer 1 Report
The MS entitled "NF-κB Signaling in Glioblastoma: Pathogenesis and Therapeutic Implications" by Shi et al. provides an overview of the role of NF-κB signaling in GBM. The article also highlights potential therapeutic approaches that target NF-κB signaling for the management of GBM, including traditional Chinese medicine substances. Overall, while the article attempts to highlight the molecular mechanisms responsible for the activation of NF-κB in GBM, it falls short in terms of organization of context, supporting evidence and critical analysis. Improving these parts should further strengthen the manuscript.
Major remarks:
-
In chapter 2 "NF-kB", a major aspect of NF-kB pathway activation - NFkB translocation to the nucleus (and export upon inactivation) is not sufficiently described. Furthermore, there is a lack of description regarding NF-κB target genes in GBM cells that change their expression upon NF-kB pathway activation.
-
The MS lacks a clear structure and organization within subparagraphs of Chapter 3, making it difficult for readers to follow the presented information. It would be helpful to use a common scheme to describe the role of the NF-kB pathway in different processes in GBM, such as proliferation and inflammation, rather than citing all the known effectors of NF-kB pathway and listing a multitude of findings related to the activation NF-κB signaling without actually explaining how activated NF-kB pathway contributes to the discussed process in GBM (proliferation, invasion, etc.). the authors should clearly explain how the activated NF-kB pathway contributes to each discussed process.
-
The manuscript lacks critical analysis and interpretation of the findings. Including a more insightful discussion would greatly benefit the manuscript. Additionally, the authors could explore and explain the potential reasons for different effects resulting from the activation of the NF-kB pathway.
-
To enhance understanding, it would be valuable to include a figure summarizing the role of NF-κB signaling in GBM.
-
It is not clear what the authors meant by "Testing index"in Table 1.
Minor remarks:
The authors should re-read the paper for some typos. The examples include:
L. 91 - "nterleukin"
L. 382 "asdas"
L. 56 - please, reformulate since it is unclear what is meant by transription factor: one transcription factor or transcription factor family
The authors should re-read the paper for some typos
Author Response
对作者的意见和建议
The MS entitled "NF-κB Signaling in Glioblastoma: Pathogenesis and Therapeutic Implications" by Shi et al. provides an overview of the role of NF-κB signaling in GBM. The article also highlights potential therapeutic approaches that target NF-κB signaling for the management of GBM, including traditional Chinese medicine substances. Overall, while the article attempts to highlight the molecular mechanisms responsible for the activation of NF-κB in GBM, it falls short in terms of organization of context, supporting evidence and critical analysis. Improving these parts should further strengthen the manuscript.
In chapter 2 "NF-kB", a major aspect of NF-kB pathway activation - NFkB translocation to the nucleus (and export upon inactivation) is not sufficiently described. Furthermore, there is a lack of description regarding NF-κB target genes in GBM cells that change their expression upon NF-kB pathway activation.
Answer: Thank you for your suggestion. We have already supplemented it chapter 2.
The MS lacks a clear structure and organization within subparagraphs of Chapter 3, making it difficult for readers to follow the presented information. It would be helpful to use a common scheme to describe the role of the NF-kB pathway in different processes in GBM, such as proliferation and inflammation, rather than citing all the known effectors of NF-kB pathway and listing a multitude of findings related to the activation NF-κB signaling without actually explaining how activated NF-kB pathway contributes to the discussed process in GBM (proliferation, invasion, etc.). the authors should clearly explain how the activated NF-kB pathway contributes to each discussed process.
Answer: Thank you for your suggestion. We have made structural adjustments.In the past two years, researchers have mostly studied the expression of genes leading to the activation of NF-κB in cells, and through some phenomenon experiments, they have concluded that the gene regulates GBM proliferation, immunity, migration and invasion, drug resistance or apoptosis through NF-κB signaling.We searched for NCBI using the keywords NF-κB and GLIOMA, with a total of 1258 articles. There is also very little research on how NF-κB signals participate in different processes. In addition, the mechanisms of occurrence and development of different tumors may vary. Before there is definite research to prove it, we will explain in detail how NF-κB activation is involved in different processes. Sorry, we are unable to resolve your suggestion.
The manuscript lacks critical analysis and interpretation of the findings. Including a more insightful discussion would greatly benefit the manuscript. Additionally, the authors could explore and explain the potential reasons for different effects resulting from the activation of the NF-kB pathway.
Answer: Thank you for your suggestion. We have supplemented it in the fourth part.
To enhance understanding, it would be valuable to include a figure summarizing the role of NF-κB signaling in GBM.
Answer: This review aims to summarize the research progress in the past two years, which mainly involves gene expression inducing activation of NF-kB signals or drug induced inactivation of NF-kB signals. However, these studies have basically not directly regulated the modification or expression of factors related to the NF-kB pathway, so we cannot present them reasonably with the figure. Secondly, this review belongs to a minor review, and there is a quantity requirement for the images and tables in the article. Therefore, we did not add any additional images.
It is not clear what the authors meant by "Testing index"in Table 1.
Answer: Thank you for your question. The meaning of this testing index is to explain which proteins are used in this study to demonstrate the activation of NF-kB signaling in GBM cells, and also to provide an intuitive detection method for future researchers who review this review.
Minor remarks:
The authors should re-read the paper for some typos. The examples include:
- 91 - "nterleukin"
- 382 "asdas"
- 56 - please, reformulate since it is unclear what is meant by transription factor: one transcription factor or transcription factor family
Answer: Thank you for your suggestion. We have carefully read and corrected the relevant errors.

Reviewer 2 Report
I enclose the file with my review below

Author Response
Thank you very much for your approval. We have optimized the manuscript, could you please guide us again.

Round 2
Reviewer 1 Report
The manuscript by Shi et al has addressed the issues that I raised before.